# Effects of 3 mg/kg Body Mass of Caffeine on the Performance of Jiu-Jitsu Elite Athletes

**DOI:** 10.3390/nu14030675

**Published:** 2022-02-05

**Authors:** María Merino-Fernández, Verónica Giráldez-Costas, Jaime González-García, Jorge Gutiérrez-Hellín, Cristina González-Millán, Michelle Matos-Duarte, Carlos Ruiz-Moreno

**Affiliations:** 1Health Science Faculty, Francisco de Vitoria University, Ctra. Pozuelo-Majadahonda Km 1800, 28223 Madrid, Spain; m.merino.prof@ufv.es (M.M.-F.); jorge.gutierrez@ufv.es (J.G.-H.); 24crgonzalez9@gmail.com (C.G.-M.); michelle.matos@ufv.es (M.M.-D.); 2Exercise Physiology Laboratory, Camilo José Cela University, 28692 Madrid, Spain; vgiraldez@ucjc.edu (V.G.-C.); jaime33gonzalez@gmail.com (J.G.-G.)

**Keywords:** fight, caffeine anhydrous, judo performance, elite athletes

## Abstract

The effects of caffeine were investigated in judo, boxing, taekwondo and Brazilian jiu-jitsu. However, this substance was never investigated regarding traditional jiu-jitsu. Therefore, the aim of this research was to analyze the effects of caffeine in the Special Judo Fitness Test (SJFT) and technical variables during combat in traditional jiu-jitsu elite athletes. **Methods:** Twenty-two young professionals of traditional jiu-jitsu, 11 men and 11 women (age = 22 ± 4 (18–33) years, body mass = 66.6 ± 10.8 (46.2–86.1) kg, height = 1.70 ± 0.9 (1.55–1.85) m) with 15 ± 7 years of experience in traditional jiu-jitsu, participated in a double-blind, counterbalanced, crossover study. In two different conditions, the traditional jiu-jitsu athletes ingested 3 mg/kg body mass of caffeine or a placebo. After 60 min, they performed the SJFT test to measure throwing performance, and subsequently, combat to analyze offensive and defensive hitting techniques. **Results:** Caffeine had a main effect on the number of throws during the SJFT test (P < 0.01). In addition, it was effective in sets 2 (13 ± 2 vs. 14 ± 2; *p* = 0.01) and 3 (12 ± 2 vs. 13 ± 1; *p* = 0.03). There was also a main effect during the test on heart rate when caffeine was ingested (F = 12.48, *p* < 0.01). The effects of caffeine were similar compared to the placebo condition regarding performance during combat both in offensive and defensive fighting variables **Conclusions:** the pre-exercise ingestion of 3 mg/kg body mass of caffeine increased performance in the SJFT test, decreased fatigue perception, and increased power and endurance perception in professionally traditional jiu-jitsu athletes. However, it did not seem to improve offensive and defensive technical actions during combat.

## 1. Introduction

Caffeine is a psychoactive substance that can be found naturally in tea leaves, cocoa, coffee, and yerba mate. Additionally, it is a substance that can be found marketed in different products such as medicines, energy drinks, or cosmetics [1]. Given its easy accessibility, caffeine is a recurrent strategy in sports due to its ergogenic characteristics. Thus, in a study that investigated the prevalence of caffeine in the urine of Olympic athletes from 2004 to 2015, three-quarters of them had consumed it prior to the competition [2]. In this context, caffeine consumption in combat sports, such as boxing or judo, has increased over the years [2]. Note that in combat sports, both in specific tests and in simulated combat, overall, the administration of caffeine between 3–9 mg/kg of the body and in different forms has shown ergogenicity [3,4,5,6]. To date, such effects have reported benefits in different combat sports, such as judo, taekwondo, Brazilian jiu-jitsu, and boxing; however, it has not yet been investigated in disciplines such as traditional jiu-jitsu [7].

Combat sports are considered intermittent disciplines mixing aerobic and anaerobic efforts, combining high intensity actions with lower intensity movements [8]. Therefore, it has been demonstrated that combat sports utilize both oxidative and non-oxidative metabolism [9,10]. Interestingly, the effect of caffeine in such disciplines has shown, in much of the evidence, an increase in performance, primarily through increased glycolytic dependence detected in response to an increase in the blood lactate concentration [7]. Such effects are produced by the main mechanism of caffeine blocking adenosine receptors. This effect on the central nervous system (CNS) generates a facilitating scenario for decreasing fatigue, a hypoalgesic effect, and also increasing the arousal levels, thus increasing sports performance [11,12].

Regarding the specific context of combat sports, caffeine ingestion has reported several benefits in both physical and technical variables [13]. In a systematic review, authors analyzed the effects of caffeine in different combat sports disciplines [7], concluding that caffeine increased the hits and performance in specific judo tests such as the special judo fitness test (SJFT) and increased throws, associated with a reduction in the perception of fatigue [14,15]. Furthermore, the ergogenic effects of caffeine are also evident in the technical actions of the combat sport analyzed. There is evidence of an increase in the technical actions measured by the number of hits when combat athletes (boxing, taekwondo, Brazilian jiu-jitsu) consumed caffeine before the simulated combat tests [3,5,6]. Technical analysis by video is a good strategy to measure fight skills, increase knowledge about combat disciplines, and provide useful information to coaches and physical trainers [16,17]. In fact, several studies have used these analyses, reporting the benefits of caffeine in combat disciplines. An increase was reported in the numbers of technical actions in several combat sports, analyzed with a camera, during simulated combats [7,18,19]. However, to the authors’ knowledge, no investigative approach collects information during matches in a real context in combat disciplines with athletes supplemented with caffeine. Given this background, caffeine could have ergogenic effects in different combat sports, but to the authors’ knowledge, there is no information regarding the ergogenicity of this substance in the traditional jiu-jitsu discipline.

Note that combat sports account for 20% of the gold medals won in the Olympic Games [20]. Thus, analysis of ergogenic supplementation such as with caffeine would help to understand the effect on different combat sports. To date, such effects have never been investigated in traditional/Japanese jiu-jitsu. Therefore, the aim of this research was to analyze the effects of caffeine on the SJFT and offensive and defensive technical actions during combat in a real context in traditional jiu-jitsu elite athletes.

## 2. Materials and Methods

Twenty-two young professionals of traditional jiu-jitsu (11 men and 11 women): 2 participants were 18 years old, 6 were 19 years old, 7 athletes were 20 years old, 2 were 21 years old, 1 was 22 years old, and the remaining 4 participants were more than 22 years old (age = 22 ± 4 (18–33) years, body mass = 66.6 ± 10.8 (46.2–86.1) kg, height = 1.70 ± 0.9 (1.55–1.85) m). Sixteen jiu-jitsu fighters were categorized in the elite junior category, and six participants were elite professionals with a certificate issued by the CSD (High Council for Sport). All the athletes were active during the 2020–2021 season. All the participants had at least seven years of experience and were training on average for 10 h per week. The participants were recruited through a search of the CSD’s public database. All the participants reported no caffeine intake 24 h prior to the study. All were informed of the procedures to be followed and the measurements to be taken, as well as the possible risks and benefits of caffeine intake. In addition, they signed the informed consent form to participate in the research (the consent and the procedures were approved by the Ethics Committee of the Francisco de Vitoria University, according to the latest version of the Declaration of Helsinki).

**Experimental Design**: For our study, we used a double-blind, randomized, placebo-controlled trial; In each experimental trial, the athletes ingested a capsule with an individualized dose of 3 mg of caffeine per kg of body mass (caffeine anhydrous powder, HSN RAW) or an identical capsule filled with cellulose. One week prior to the competition, they were asked by email to send their weight and height measurements. Subsequently, caffeine capsules were developed according to the weight sent by the athletes. The caffeine capsules were opaque, and neither the participants nor the researchers knew what they were ingesting. These capsules were identified with an alphanumeric code, and an external researcher randomized the caffeine or placebo administration. On the experimental day, the fighters went to the facilities one hour before the SJFT test. The athletes ingested the capsule 45 min prior to the warm-up. After 10 min of warm-up, they went to the tatami, where they performed the SJFT test followed by traditional jiu-jitsu combat. This procedure was repeated after one week of carrying out the scheduled bouts (all of them recorded for subsequent analysis by experts).

The SJFT test is considered the gold standard test specific to judo grip skills. The SJFT was divided into three consecutive rounds: the first round with 15 s (s) duration followed by two rounds of 30 s duration. The breaks between the first and the second rounds were 10 s in duration. In each round, researchers counted the number of throws in both conditions. The designated jiu-jitsu fighter (Tori) performed the highest number of throws of the judo technique called Ippon-seoinage. Two designated partners (Uke) with similar body mass received the ippons for the test to be performed. The participants’ Uke were the same in both conditions. To standardize the test across conditions, the Toris were placed 6 m apart, and the Uke was to run from Tori#1 to Tori#2 as fast as possible, to be thrown using the Ippon-seoinage technique, then continue to run from Tori#2 to Tori#1 to repeat the running plus throwing sequence as many times as possible during the time of each round. SJFT performance was measured by the number of complete and valid throws made in the three rounds, which were confirmed in real-time by two independent judges experienced in refereeing this test; they were blinded to the treatments. The sum of total throws completed in two SJFTs performed in the investigation was also calculated as a measure of performance. Throughout the entire test, heart rate was monitored with a Pellor stopwatch with an LED display, as well as heart rate monitors placed on the chest. In addition, the SJFT index for each trial was calculated for each participant in the SJFT test as follows [21]: index%=(Heart rate final+heart rate 1minuteafter the test) ÷number of throws . At the end of the SJFT test, the athletes were asked about their perceived endurance, strength, and fatigue by means of a questionnaire. The questionnaire included a numerical scale from 1 to 10 points. Before starting to answer it, the athletes were informed that 1 point meant the minimum amount of that item and 10 points meant the maximum amount [22].

During combat, athletes performed between 3 and 5 bouts each per day, both days with the same partner (similar competition weights). Participants competed in one-on-one combats, and both fighters were evaluated with the official rules of traditional jiu-jitsu. Combats were recorded with two cameras placed at different angles and were directed by a referee to simulate as real a fight as possible. Traditional jiu-jitsu combats are composed of two parts: the first part includes techniques using hits with punches and kicks (atemis), the second part includes takedowns, projections, dislocations, and strangulations. The technical actions of striking during the first half were counted through the visualization of the fights. In the second and third parts, the technical actions of groundwork and projections were counted, differentiating techniques with low score possibility and techniques with high score possibility. There was a rest of 10 min between combat parts. Technical skills measured were (i) *action technique without score*—Athlete performs a technical action with the fist or leg (atemi waza), there is no contact (ii) *blocked technique*—the athlete performs an atemi waza action (punch or kick) and is blocked by the opponent (iii) *technique with low score possibility*—the athlete tries to throw the opponent but does not succeed and causes no imbalance (iv) *technique with high score possibility*—the athlete tries to throw the opponent (nage waza) with good imbalance and almost scores a point, but the opponent manages to defend himself. Researchers used two cameras (Handycam HDR-XR200VE, Sony, Spain) located contiguously on two sides of the tatami, with a wide view of the combat zone. For the analysis, a blinded researcher specialized in the sport, visualized the videos and counted all the technical actions.

**Statistical Analysis:** The study data were blindly entered into the JAMOVI package [23] and subsequently analyzed. The Shapiro–Wilk test was the first statistical analysis to confirm the normality of the quantitative variables. Consequently, parametric statistical tests were performed for the two conditions. The SJFT test totals and the study variables for the jiu-jitsu technical actions were tested using Student’s t-test for pairwise comparisons. A two-way repeated-measures ANOVA was used to analyze performance between the series during the SJFT test. For all statistical tests, the significance level was *p* < 0.05. The data is presented as mean ± standard deviation. Figure 1 depicts by COMSORT flow diagram the experimental design and methods

## 3. Results

Figure 2 depict similar results in set 1 for the number of throws (7 ± 1 vs. 8 ± 1; *p* = 0.38), increasing by 1.3%, and heart rate (102 ± 16 vs. 108 ± 18; *p* = 0.16) enhancement of 3.5% when caffeine was administered compared to the placebo in the SJFT test. On the contrary, regarding throws in the SJFT test, caffeine was significantly different compared to the placebo condition in sets 2 (13 ± 2 vs. 14 ± 2; *p* = 0.01) and 3 (12 ± 2 vs. 13 ± 1; *p* = 0.03), increasing 8.1% and 5.6% respectively. Regarding heart rate, at the end of the SJFT set 2, it increased by 3.2% (173 ± 9 vs. 179 ± 10; *p* = 0.04) and in set 3 showed an enhancement of 3.1% (161 ± 10 vs. 166 ± 7; *p* = 0.04) when athletes ingested caffeine. By means of the ANOVA caffeine showed a significant effect on SJFT performance (F = 9.34, *p* < 0.01) and heart rate (F = 12.48, *p* < 0.01).

Caffeine had significant effects compared to the placebo condition regarding total throws and in the index and average heart rate during the SJFT test. With respect to the perceptual variables at the end of the SJFT test, caffeine significantly increased strength perception and endurance perception and reduced fatigue perception (Table 1).

Caffeine did not modify the number of offensive hits as *action techniques without score* and defensive techniques as *blocked techniques*. On the other hand, caffeine presented very similar results with caffeine ingestion compared to the placebo condition in *techniques with low score possibility* and *techniques with high score possibility* (Table 2).

## 4. Discussion

The purpose of this study was to analyze the effects of caffeine on the SJFT and offensive and defensive technical actions during combat, with a real context in professional traditional jiu-jitsu athletes. For this purpose, two researchers administered 3 mg/kg of body mass of caffeine on two different days. To the authors’ knowledge, this is the first research to study the effects of acute caffeine intake in traditional jiu-jitsu fighters in the SJFT test and during real combat. The main findings were that: (i) caffeine increases SJFT test performance, heart rate, and strength and endurance perception, and diminishes fatigue perception (ii) caffeine did not modify the number of either offensive or defensive technical actions during jiu-jitsu traditional combat in a real context.

To date, the SJFT test is a gold standard field test to evaluate specific performance in judo [21]. Since traditional jiu-jitsu has a judo element, it was used to evaluate the performance of fighters. As demonstrated in Figure 2, caffeine was effective for sets 2 and 3, while set 1 remained similar in both conditions. These results are evidenced in several investigations where the effectiveness of caffeine, administered in different doses and forms, seems to help increase performance in this test [24,25,26]. Although there is old research showing that the ability of caffeine in doses of 200–900 mg to improve hand-eye performance could negatively influence fine motor coordination [27], the increase in performance in this test could be due to the main mechanism of caffeine. By blocking adenosine receptors in the brain A_1_ and A_2_, it creates a favorable situation for delaying fatigue during exercise. Moreover, due to the secretion of catecholamines, caffeine produces a hypoanalgesic effect during exercise, and, as a consequence, the perception of fatigue is reduced compared to the placebo [14]. Therefore, our data suggest that caffeine is effective in this test, at least statistically, even though, in the first set, the results were similar, probably due to the main physiological mechanism. Additionally, the increase in perceived strength and endurance in this test cooperates to increase levels of excitability because caffeine is a CNS stimulant [14].

Regarding the technical variables, our results were similar when the caffeine supplementation was compared to the placebo condition. Compared with our results, on the contrary, it was demonstrated in different investigations that the technique (the number of hits) would increase during simulated combat when caffeine was ingested before the fight [7]. In a study where Brazilian jiu-jitsu discipline was investigated, technical actions increased when fighters ingested caffeine during simulated combat [3]. The same occurred in another sport combat such as fighting, where the number of hits increased during simulated boxing combat, mainly in the second round [6]. However, similar to our study, caffeine did not improve technical actions during a taekwondo match [5]. Furthermore, similar outcomes in both conditions could be conditioned by the randomization of caffeine administration. Methodologically, caffeine was administered on a randomized basis to both fighters, who were the same as those who competed in the two fights. Thus, it could have been conditioned due to the randomization of both caffeine conditions in the same real combat (i.e., caffeine vs. caffeine). Therefore, the results obtained during the combat could benefit both offensive and defensive technical skills. This may be justified by a study where the methodology was similar, where they evaluated fights in a simulated competition context and found that there were no differences in the technical actions [19]. Therefore, to the authors’ knowledge, caffeine could also improve defensive performance when this dependent variable is accounted for.

This research has several limitations that should be discussed: (i) the study design is lacking in its randomization of caffeine administration because, in the specific jiu-jitsu traditional combat, both athletes could be supplemented with the substance. This would lead to similar results in the offensive and defensive dependent variables; (ii) this combat discipline is new in science; therefore, the methodology is conditioned to the regulations of the sport itself; and (iii) finally, no blood lactate tests were taken; therefore, we have no data on the physiological variables increasing glycolytic dependence associated with caffeine and the performance of both tests.

## 5. Conclusions

From a scientific perspective, the administration of caffeine in combat sports appears to be beneficial for performance enhancement [7]. The results suggest that when 3 mg/kg of body mass were administered, there was an increase in performance in the SJFT test, decreased fatigue perception, and increased power and endurance perception. However, it did not seem to improve offensive techniques during real combat. Despite these results, investigations related to this combat sports discipline should be increased to obtain stronger conclusions. Nevertheless, from a practical perspective, the authors recommend the administration of this substance to improve performance because of the benefits with respect to decreased fatigue and increased performance in grip variables (SJFT), although it should be further recommended by the nutritionist or trainer.

## Figures and Tables

**Figure 1 nutrients-14-00675-f001:**
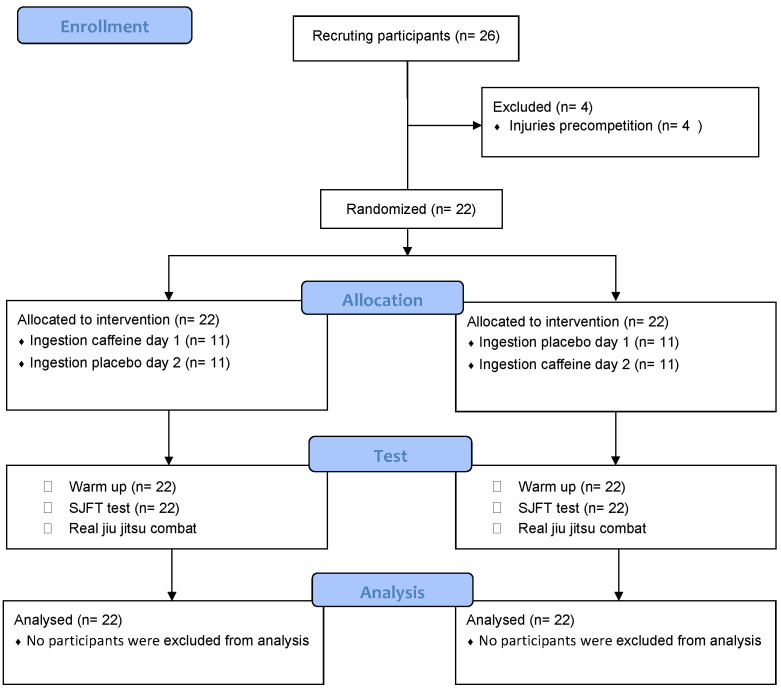
CONSORT flow diagram of methodology and data analysis.

**Figure 2 nutrients-14-00675-f002:**
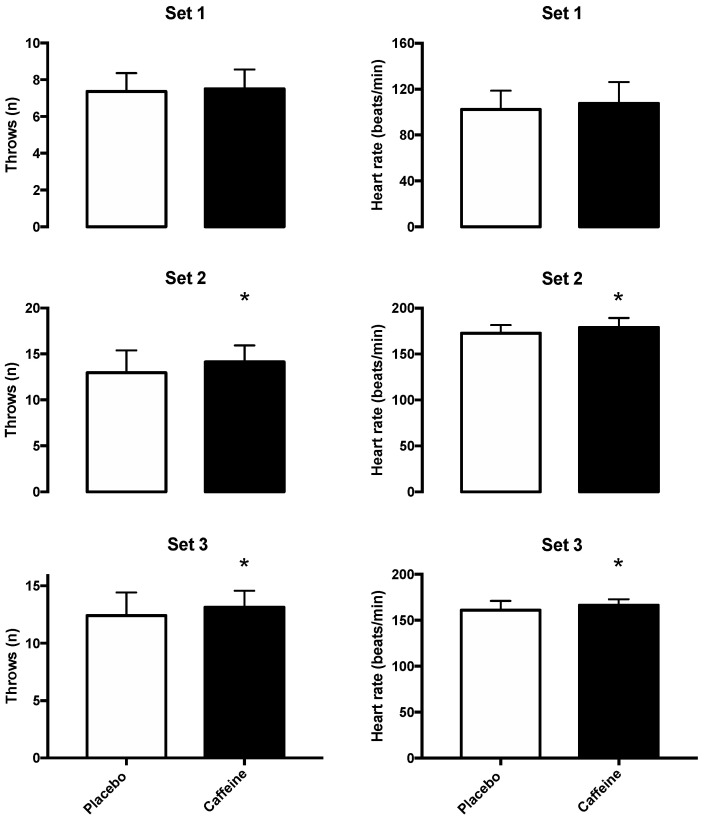
Number of throws and heart rates during the different series in the SJFT test after the administration of 3 mg/kg body mass caffeine and placebo. Data are shown as mean ± SD for 22 athletes. (*) Caffeine is significantly different from the placebo condition (*p* < 0.05). Left graphs show the number of throws, and right graphs show the average heart rate (bpm). White columns are placebo conditions, and black columns are caffeine conditions.

**Table 1 nutrients-14-00675-t001:** Total number of throws, work index, heart rate and perceptual variables obtained after the SJFT test with the administration of 3 mg/kg body mass of caffeine and placebo.

Variables (Units)	Placebo	Caffeine	*p*-Value
Throws (n)	33 ± 5	35 ± 4	0.01
SJFT index (%)	9.7 ± 1.7	10.3 ± 1.3	0.04
Heart rate (bpm)	145 ± 7	151 ± 7	<0.01
Strength perception (a.u)	5.8 ± 1.7	6.8 ± 1.6	<0.01
Endurance perception (a.u)	5.7 ± 1.7	6.7 ± 1.4	0.02
Fatigue perception (a.u)	6.7 ± 1.7	5.4 ± 2.0	0.02

Data is shown as mean ± SD for 22 athletes. Caffeine different to placebo condition (*p* < 0.05).

**Table 2 nutrients-14-00675-t002:** Total offensive and defensive technical actions during traditional jiu-jitsu combat after the administration of 3 mg/kg body mass of caffeine or placebo.

Variables (Units)	Placebo	Caffeine	*p*-Value
Action technique without score (n)	2.0 ± 1.2	1.7 ± 0.8	0.31
Blocked technique (n)	3.2 ± 2.8	3.7 ± 2.3	0.41
Technique with low score possibility (n)	1.6 ± 1.0	1.5 ± 1.0	0.71
Technique with high score possibility (n)	1.0 ± 0.6	1.2 ± 1.1	0.32

Data are shown as mean ± SD for 22 athletes. Caffeine different to placebo condition (*p* < 0.05).

## Data Availability

Not applicable.

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
