# Peer review of "Effects of 3 mg/kg Body Mass of Caffeine on the Performance of Jiu-Jitsu Elite Athletes"

_nutrients, 2022, doi:10.3390/nu14030675_

Round 1

Reviewer 1 Report

It is a good and interesting study.

There is great need for the lenguage correction! Even the title is incorrect! There are many grammatic errors in the text. Even the absract needs a lot of improvement.

Materials and methods are well writen, the study is well designed, data are interesting. The Results section can be improved,Table 1 is not understandable. Fatigue preception is 5+/_175? I dont see it possible.

The Discussion section needs improvement.

Author Response

We sincerely thank the expert Reviewers and the Guest Editor for carefully proofreading the manuscript and for their helpful and constructive comments. We have addressed all the points raised by the Reviewers in this response letter and we have highlighted any changes to manuscript in red. We believe that our manuscript has been improved by the suggested changes.

Reviewer #1

Comments and Suggestions for Authors

It is a good and interesting study.

There is great need for the lenguage correction! Even the title is incorrect! There are many grammatic errors in the text. Even the absract needs a lot of improvement.

Thank you for your comment. We have changed the title to make to clearly indicate the content of the investigation. Now the title reads:

 “Effects of caffeine in performance of jiu-jitsu elite athletes”.

Materials and methods are well written, the study is well designed, data are interesting. The Results section can be improved, Table 1 is not understandable. Fatigue preception is 5+/_175? I dont see it possible.

Thank you for your comment. We have changed the typographical error in table 1.

The Discussion section needs improvement.

Thank you for your contribution. We have completed with more information discussion section.

Reviewer 2 Report

Overall I believe this study has merit however, there are several missing pieces of information. Additionally, I would recommend that the authors thoroughly proof-read the manuscript to ensure that they have caught all writing errors. 

Introduction

Overall, good job writing the introduction however there are several citation errors. For example when you report the use of a systematic review, you cite 2 different sources.

I appreciate the authors writing a succinct objective statement however, as a reader I would also appreciate the authors providing information as to how technical offensive and defensive skills are measured in real combat situations. The authors should spend some time in the introduction talking about how these outcomes can be measurable and how the authors plan to measure them in their manuscript.

Methods

The methodology needs significantly more information. Below are my point by point comments

  1. What was the inclusion/exclusion criteria
  2. What is the overall age distribution? 
  3. What is the overall experience distribution?
  4. I think you meant to write .9m for SD for height? and if so, that is a VERY large variation in height
  5. Seems as if there was a large variation in weight as well
  6. Based on large variations in experience, age, body mass and height, I would recommend reporting range and median.
  7. How were participants recruited? 
  8. How many potential participants were recruited? How many qualified? 
  9. Was there an a priori power analysis completed prior to recruitment? If so, how?
  10. I am assuming this was a double-blinded, randomized placebo controlled cross-over design based on the entire description?
  11. How was it double-blinded if the ingested caffeine was 3mg/kg for body mass? It is possible, but please explain how
  12. How were the allocation days randomized?
  13. What was the subjective effort questionnaire? Was it simply a report of RPE? If so, what RPE scale did you use? Please dedicate a section to describe the subjective effort questionnaire
  14.  Great job explaining the SJFT. Lines 95-97 is a run on and I had a very difficult time comprehending it. I had to re-read it multiple times. 
  15. Where were the cameras placed when recording combat?
  16. Line 114-115, you didn't complete your thought
  17. Did you control for prior night's sleep? 
  18. In the statistical analysis section you state that non-parametric statistics were performed for the two conditions however, you state that the Student's T-test was performed for pairwise comparisons. Additionally, you report using a two-way repeated measures ANOVA. Both of these tests are parametric statistical analyses. I understand that there are no non-parametric equivalents to the two-way repeated measures ANOVA however, if the data is non-parametric shouldn't you use a Mann-Whitney U?
  19. Did you run an analysis looking at differences by day before examining differences by caffeine ingestion? It might help you determine whether there were learning effects or whether the results that you report were because of caffeine. 

Results

I have several comments regarding the results section

  1. Please provide a CONSORT flow diagram. You may also provide this in the methods section if you'd like.
  2. What does the percentage at the end of the reported results indicate/measure?
  3. There are several grammatical errors in the reporting of the results
  4. For the tables, please be careful where the decimals are placed. Also please standardize how many significant digits you present

Discussion

Below are my comments regarding the discussion

  1. The first paragraph in the discussion section in filled with run-ons.
  2. To explain the results of the SJFT gold standard field test, you may want to examine the work by Jacobson and colleagues which reports that doses such as the one that your participants had improves hand-eye coordination. This may have also played a role in explaining your results. Fuller and colleagues also report that some people have an improvement in fine motor task performance with caffeine consumption
  3. In lines 198-208 you make some very interesting observations. I think you might be on to something. 
  4. Regarding your limitation ii, did you try to run an analysis by day rather than by intervention? This could help you determine whether there was a learning effect or not. I would recommend using this technique in your statistical analysis to determine if there were any learning effects.

Author Response

Reviwer 2

Overall I believe this study has merit however, there are several missing pieces of information. Additionally, I would recommend that the authors thoroughly proof-read the manuscript to ensure that they have caught all writing errors.

Thank you for this positive comment. We have amended the manuscript to solve the issues that you have pointed out. We hope that this version satisfies the high-level criteria for publication in Nutrients.

Introduction

Overall, good job writing the introduction however there are several citation errors. For example when you report the use of a systematic review, you cite 2 different sources.

Thank you for this comment. We have removed the citation that did not apply to the phrase

I appreciate the authors writing a succinct objective statement however, as a reader I would also appreciate the authors providing information as to how technical offensive and defensive skills are measured in real combat situations. The authors should spend some time in the introduction talking about how these outcomes can be measurable and how the authors plan to measure them in their manuscript.

Thank you for this comment. We have included this text in the article.

Technical analysis by video, is a good strategy to measure fight skills and increases knowledge about combat discipline and could have util information to coaches and physical trainers (16,17). In fact, several studies have used these analyses reporting benefits of caffeine in combat disciplines. Despite, have been reported increased in the numbers of throws in jiu jitzu Brazilian analysed with a camera (18,19). However, to authors knowledge, no investigation approach information during real fight in combat sport discipline. Given this background, caffeine could have ergogenic in different combat sports, but to the authors' knowledge, there is no information regarding the ergogenic of the substance in traditional jiu-jitsu discipline.

Methods

The methodology needs significantly more information. Below are my point by point comments

What was the inclusion/exclusion criteria

Thank you for your comment. In the inclusion criteria, they had to be athletes with the qualification of high active level, or be high performance athletes during the 2019-2020 season.

16 jiu jitsu fighters were categorized in the elite junior category, and six participants were elite professional with a certificate issued by CSD (Superior Council of Sports). Athletes were active during 2020-2021 season. All participants had, at least, seven years of ex-perience, and they were training on average for 10 hours per week. Participants were recruited through a search of the CSD's public database

What is the overall age distribution?

Thank you for your comment. Except for 4 participants who were more than 22 years old, 2 participants were 18, 6 were 19, 7 were 20, 2 were 21, and 1 was 22.

2 participants were 18 years old, 6 were 19 years old, 7 athletes were 20 years old, 2 were 21 years old and 1 was 22, remaining 4 participants were more than 22 years old

What is the overall experience distribution?

Thank you for your comment. All the athletes had at least 7 years of experience training the traditional jiu jitzu discipline. And they trained an average of 10 hours a week.

All participants had, at least, seven years of experience, and they were training on average for 10 hours per week

I think you meant to write .9m for SD for height? and if so, that is a VERY large variation in height

Thank you for this comment. This was an error on our part and we have corrected

Seems as if there was a large variation in weight as well

Thank you for this comment. The data seems to have a large variation, because the overall sample of athletes is divided between both sexes.

Based on large variations in experience, age, body mass and height, I would recommend reporting range and median.

Thank you for your observation. We included the range of this descriptive variables

How were participants recruited?

Thank you for your comment. The authors, we were searching in public database of the website of the CSD (Consejo Superior de deporte in spain) to see the athletes who are DAN of jiu jitsu. Once we found them, we made a search filtering because they are registered as judocas. Afterwards, we called the Spanish traditional Jiu Jitsu federation to analyze which athletes had the high performance requested and approved that year. Finally, we contacted the clubs to which they belonged, making the request.

All athletes were recruited by means of a search in the public database of the website of the CSD. Participants reported that they had no injuries, were not on prescription medications or supplements for the duration of the study, were nonsmokers, and were classified as light caffeine users based on dietary reports

How many potential participants were recruited? How many qualified?

Thank you for your comment. There were 26 participants recruited. However, there was a sample mortality of 4 due to pre-competition injuries. They didn’t participate in a competition. The combats were with a real context.

Was there an a priori power analysis completed prior to recruitment? If so, how?

Thank you for this comment. We didn´t perform a priori sample size.

I am assuming this was a double-blinded, randomized placebo controlled cross-over design based on the entire description?

Thank you for your comment. The description of experimental design is in line 85-86

How was it double-blinded if the ingested caffeine was 3mg/kg for body mass? It is possible, but please explain how

Thank you for this comment. The days prior to the data collection, they were sent by email to be weighed and measured. Subsequently, caffeine pills were made according to the weight sent. The caffeine pills were opaque and neither the participants nor the researchers knew what they were ingesting.

One week prior to the competition, they were sent by email to be weighed and measured. Subsequently, caffeine capsules were made according to the weight sent by athletes. The caffeine capsules were opaque and neither the participants nor the researchers knew what they were ingesting. These capsules were identified with a alphanumeric code. One week prior to the competition, they were sent by email to be weighed and measured. Subsequently, caffeine capsules were made according to the weight sent by athletes. The caffeine capsules were opaque and neither the participants nor the researchers knew what they were ingesting. These capsules were identified with a alphanumeric code. Ex-perimental day, fighters went to facilities one hour before to the SJFT test. 45 min prior to warm up athletes ingested the capsule. After 10 min to warm-up, they were presented to the tatami where they performed the SJFT test followed by the jiu jitsu traditional combat for the fighters. This procedure was repeated after one week carried out the scheduled bouts (all of them recorded for subsequent analysis by experts).

How were the allocation days randomized?

Thank you for your comment. A researcher external to the study performed the randomization, dividing and classifying them with an alphanumeric code for subject identification.

an external researcher randomized the caffeine or a placebo administration.

What was the subjective effort questionnaire? Was it simply a report of RPE? If so, what RPE scale did you use? Please dedicate a section to describe the subjective effort questionnaire

Thank you for this comment. The questionnaire was based on the article by Salinero JJ, Lara B, Abian-Vicen J, Gonzalez-Millán C, Areces F, Gallo-Salazar C, et al. The use of energy drinks in sport: perceived ergogenicity and side effects in male and female athletes. Br J Nutr. 2014;

At the end of the SJFT test, the athletes were asked about their perceived endurance, strength, and fatigue by means of a questionnaire. The questionnaire included a nu-merical scale from 1 to 10 points. Before starting to answer it, the athletes were informed that 1 point meant the minimum amount of that item and 10 points meant the maximum amount (21).

 Great job explaining the SJFT. Lines 95-97 is a run on and I had a very difficult time comprehending it. I had to re-read it multiple times.

Thank you for the comment. Authors change the phrase for a better understanding

the first round with 15 seconds (s) duration followed by two rounds of 30 s duration. The breaks between the first and the second rounds were 10 s in duration. In each round, researchers counted the number of throws in both conditions. Jiu-jitsu fighter denominated (Tori) performed the highest number of throws of the judo technique called Ip-pon-seoinage. Two partners denominated (Uke) with similar body mass, received the ippons for the test to be performed. The participants Uke were the same in both conditions

Where were the cameras placed when recording combat?

Thanks for the comment. Two cameras were used and placed outside tatami. They were strategically placed to be able to see all the angles of the fight.

Researchers use two cameras (Handycam HDR-XR200VE, Sony, Spain) located contig-uously on two sides of the tatami, with a wide view of the combat zone. For the analysis, a researcher specialized in the sport, visualized the videos, and blindly counted all the technical actions.

Line 114-115, you didn't complete your thought

Thank you for the insightful observation. The authors made a typographical error. It has been corrected.

Did you control for prior night's sleep?

Thank you for this insightful comment. The authors did not consider controlling for this variable. Because it was a jiu jitsu championship, we did not want to disturb the participants beforehand. However, they were recommended to rest for at least 8 hours.

In the statistical analysis section you state that non-parametric statistics were performed for the two conditions however, you state that the Student's T-test was performed for pairwise comparisons. Additionally, you report using a two-way repeated measures ANOVA. Both of these tests are parametric statistical analyses. I understand that there are no non-parametric equivalents to the two-way repeated measures ANOVA however, if the data is non-parametric shouldn't you use a Mann-Whitney U?

Thank you for this accurate comment. In the writing, there was a typographical error, made by the corresponding author. The statistics used was parametric, so, the statistical methods are the appropriate ones, as written in the text.

Did you run an analysis looking at differences by day before examining differences by caffeine ingestion? It might help you determine whether there were learning effects or whether the results that you report were because of caffeine.

Results

I have several comments regarding the results section

Please provide a CONSORT flow diagram. You may also provide this in the methods section if you'd like.

Thank you for your contribution. We have included the CONSORT flow diagram.

Figure 1. CONSORT flow diagram to methodology and data analysis

What does the percentage at the end of the reported results indicate/measure?

Thank you for the insightful comment. The authors explained in the results section the percentage change when caffeine was administered compared to placebo.

There are several grammatical errors in the reporting of the results

Thank you for the critical comment. The authors have made a extensive revision of the English language.

For the tables, please be careful where the decimals are placed. Also please standardize how many significant digits you present

Thank you for your comment and observation. The authors have corrected the typographical errors and homogenized the table.

Discussion

Below are my comments regarding the discussion

The first paragraph in the discussion section in filled with run-ons.

Thank you for your input. The authors have changed the first paragraph for a better understanding for the potential reader.

The purpose of this study was to analyze the effects of caffeine on the SJFT and technical actions, offensive and defensive, during combat with a real context in professional traditional jiu-jitsu athletes. For this purpose, two researchers administered 3 mg/kg of body mass of caffeine in two different days. To authors knowledge, this is the first research to study the effects of acute caffeine intake in traditional jiu-jitsu fighters in the SJFT test and during real combat. The main findings were that (i) caffeine increases SJFT test performance and increases heart rate and diminishes fatigue perception and increases strength and endurance perception (ii) caffeine did not modify the number, of neither offensive nor defensive technical actions during a real Jiu jitsu traditional combat with a real context.

To explain the results of the SJFT gold standard field test, you may want to examine the work by Jacobson and colleagues which reports that doses such as the one that your participants had improves hand-eye coordination. This may have also played a role in explaining your results. Fuller and colleagues also report that some people have an improvement in fine motor task performance with caffeine consumption

Thank you for your contribution. We have included this text.

Although there is old research showing that the ability of caffeine in doses (200-900 mg) to improve hand-eye performance could negatively influence fine motor coordination (27), the increase in performance on this test could be due to the main mechanism of caffeine

In lines 198-208 you make some very interesting observations. I think you might be on to something.

Thank you for this positive comment.

Regarding your limitation ii, did you try to run an analysis by day rather than by intervention? This could help you determine whether there was a learning effect or not. I would recommend using this technique in your statistical analysis to determine if there were any learning effects.

Thank you for the insightful comment. Certainly, we performed the analysis, there were no differences per day. However, being elite traditional jiu jitsu fighters, both the test and the combat, and with the large participant experience, they were familiar. Therefore, with the permission of both reviewers, I proceed to delete that sentence as a limitation of the study.

Round 2

Reviewer 2 Report

I'd like to thank the authors for addressing my concerns however, there are still significant grammatical errors in this manuscript. Additionally, the manuscript still states that nonparametric measures were performed, but does not identify which ones or for what variables. There are also significant errors in the data tables. I would highly recommend that ALL the authors proof read this work.

Author Response

I'd like to thank the authors for addressing my concerns however, there are still significant grammatical errors in this manuscript. Additionally, the manuscript still states that nonparametric measures were performed, but does not identify which ones or for what variables. There are also significant errors in the data tables. I would highly recommend that ALL the authors proof read this work.

We are sincerely grateful for the reviewer's input. We hope that we have approached all the directions. 

1. We have corrected the grammatical errors with a native English reviewer. The changes are highlighted in green.

2. We have changed the erroneous results in the table.

3. We have changed from non-parametric to parametric.